# Catalytic condensation for the formation of polycyclic heteroaromatic compounds

Daniel Forberg[1], Tobias Schwob[1] & Rhett Kempe [1]

The conservation of our global element resources is a challenge of the utmost urgency. Since aliphatic and aromatic alcohols are accessible from abundant indigestible kinds of biomass, first and foremost lignocellulose, the development of novel chemical reactions converting alcohols into important classes of compounds is a particularly attractive carbon conservation and $CO_2$-emission reduction strategy. Herein, we report the catalytic condensation of phenols and aminophenols or aminoalcohols. The overall reaction of this synthesis concept proceeds via three steps: hydrogenation, dehydrogenative condensation and dehydrogenation. Reusable catalysts recently developed in our laboratory mediate these reactions highly efficient. The scope of the concept is exemplarily demonstrated by the synthesis of carbazoles, quinolines and acridines, the structural motifs of which figure prominently in many important natural products, drugs and materials.

---

[1] Anorganische Chemie II – Katalysatordesign, Universität Bayreuth, 95440 Bayreuth, Germany. Correspondence and requests for materials should be addressed to R.K. (email: kempe@uni-bayreuth.de)

The sustainable use of the resources of our planet has become a necessity and one of the great challenges of our time. For chemistry, with its enormous demand for carbon, the move away from the currently dominating technologies consuming oil and related fossil resources toward more sustainable strategies is indispensable in the longer term. An attractive alternative carbon source, if responsibly chosen, is biomass. Lignocellulose, a class of biomass that is abundantly available, barely used and indigestible[1] can be converted to alcohols via pyrolysis and hydrogenation steps[2]. Thus, alcohols can be regarded as the sustainable alternative to oil crack products, which are the basis of many of the chemical compounds produced today. Consequently, the development of novel reactions that convert alcohols to important classes of compounds is a central topic in chemistry[3]. A concept has been introduced recently by which the combination of dehydrogenation and condensation steps permits the synthesis of important aromatic N-heterocycles, such as pyrroles[3–9], pyridines[10–13], and pyrimidines[14–16], starting from aliphatic alcohols (Fig. 1a)[17, 18]. In this concept, condensation steps deoxygenate and dehydrogenations enable aromatization. The liberation of $H_2$ in the course of these reactions, for the pyridine synthesis (Fig. 1a), for instance, three equivalents per pyridine molecule, is appealing to us. It allows for novel synthesis concepts where additional steps dovetail with the liberation of hydrogen and use it for reductive substrate activation before it is released.

Herein, we report on a concept of a sustainable synthesis in which phenols and aminophenols or aminoalcohols undergo a catalytic condensation reaction for the formation of polycyclic N-heteroaromatic compounds (Fig. 1b). This process involves hydrogenation as well as multiple dehydrogenation and condensation steps (Fig. 2a). We exemplarily applied this concept to the synthesis of carbazoles, quinolines and acridines (Fig. 1b). We first hydrogenate the phenols. In the next step, a dehydrogenation–condensation sequence is applied giving rise to polycyclic compounds combining saturated and aromatic rings. Finally, dehydrogenation leads to purely aromatic polycyclic N-heterocyclic compounds (Fig. 2a). The reaction proceeds via polycyclic pyrrole and pyridine intermediates, interesting compounds which can be isolated if desired. The overall reaction may be run without isolation of the intermediates by just adding or removing the catalyst for the anticipated reaction step. For this purpose, we use efficient reusable catalysts recently developed in our laboratory[8, 19]. The concept might be suitable to meet some challenges associated with aryl ether hydrogenolysis, a key step of lignin valorization[20]. This reaction is unselective for most of the catalysts leading to cyclohexanol products rather than to phenols[20]. The target polycyclic aromatic N-heterocycles have a wide range of conceivable applications in medicine and materials science.

## Results

**Optimization of the reaction conditions**. The simple condensation of phenols and aminophenols to carbazoles becomes explicable by presuming a reaction sequence, as shown in Fig. 2a. In the first step, the two starting phenols are hydrogenated. The resulting cyclohexanols could undergo an acceptorless dehydrogenative condensation (ADC)[3] to afford an octahydrocarbazole intermediate which, in turn, is dehydrogenated to give the final carbazole product. To run such reaction sequences

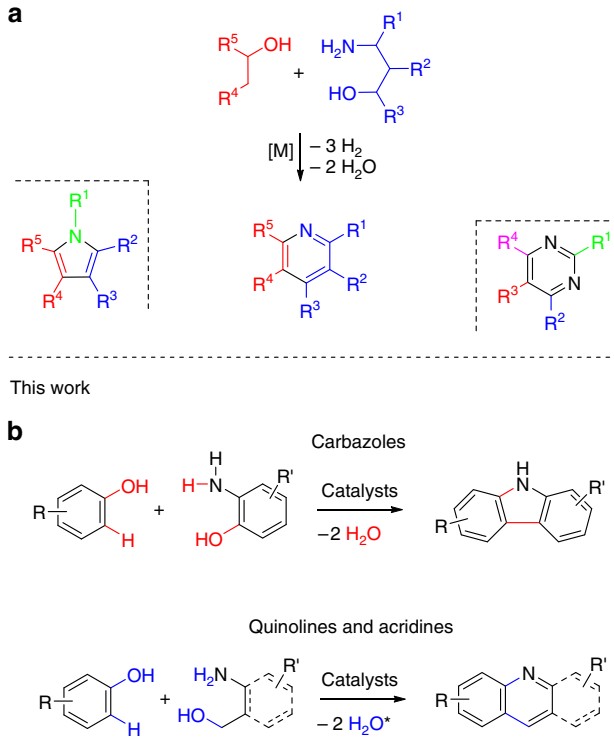

**Fig. 1** Synthesis of aromatic N-heterocycles and polycyclic heteroaromatic compounds. **a** Known sustainable two-, three- and four-component reactions linking alcohols to important aromatic N-heterocyclic compounds. The synthetic pathway is shown for the pyridine synthesis (R = substituents). **b** Catalytic reaction of phenols and aminophenols or aminoalcohols to polycyclic aromatic compounds disclosed here— synthesis of carbazoles, quinolines and acridines via catalytic condensation (*in addition, $H_2$ is liberated; acridines 1 equiv. and quinolines 2 equiv.)

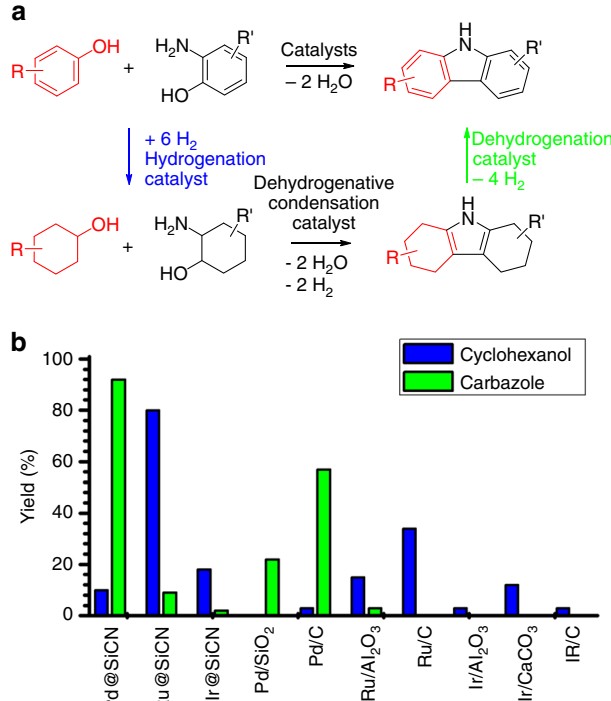

**Fig. 2** Reaction sequence and catalyst screening. **a** The condensation of phenols and aminophenols via hydrogenation and multiple dehydrogenation/condensation steps. **b** Identification of an efficient hydrogenation and dehydrogenation catalyst. Reaction conditions: hydrogenation of phenol: 1 mmol phenol, 50 °C, 3 bar $H_2$, 0.03 mol% active Ru, 1 mL $H_2O$, 5 h; dehydrogenation of octahydrocarbazole: 1 mmol substrate, 0.75 mL diglyme, 0.18 mol% active metal, 190 °C, Ar flow (4–6 mL/min)

as a hassle-free procedure without the need for isolating intermediates, easy to separate reusable catalysts would be advantageous. Based on the recently made progress in ADC reactions and the many catalysts described to mediate such reactions[3–16], we focused, firstly, on the development of efficient hydrogenation and dehydrogenation catalysts. A comparison of commercially available ruthenium (Ru), palladium (Pd) and iridium (Ir) catalysts revealed only low conversions (maximum 34 %) under the conditions given: hydrogenation of phenol at 50 °C applying 3 bar $H_2$ pressure for 5 h with a catalyst loading of 0.03 mol% active metal (Fig. 2b). Since Ru catalysts showed the highest activity and very small metal nanoparticles had previously been generated in a silicon carbonitride (SiCN) matrix[21], we used a Ru-SiCN nanocomposite catalyst (Ru@SiCN) recently developed in our laboratory.[19] This Ru@SiCN catalyst achieved 80 % conversion for the hydrogenation of phenol under the screening conditions given and was, thus, identified as the most active catalyst under scrutiny. Ru@SiCN is able to hydrogenate a variety of phenols (Table 1) quantitatively with just 0.03 mol% loading of catalytically active metal (50 °C and 20 bar $H_2$ pressure). Similar to Ru@SiCN, a palladium-SiCN nanocomposite catalyst (Pd@SiCN)[19] was identified as the most active dehydrogenation catalyst (Fig. 2b).

**Substrate scope of the catalytic condensation concept**. Having efficient catalysts for the hydrogenation and the dehydrogenation step available, we studied the scope of the novel catalytic condensation concept. We used Ir catalysts introduced by our group recently; preferentially a reusable Ir@SiCN catalyst[8] or a $PN_5P$-pincer catalysts[3] if its performance was significantly better for a certain combination of starting materials. The reaction of phenols with 2-aminophenols generates carbazoles (overall) as an example of a substance class addressable with our synthesis concept. Octahydro-1H-carbazole (1a) was isolated in a yield of 81% (Table 2, pathway I, top). Here, we observed that the homogeneous Ir-pincer catalyst **I** was more active and selective than the reusable Ir@SiCN catalyst at 140 °C. To our delight, the yields could be improved toward similar results as observed for the homogeneous Ir catalyst **I** by an increase of the reaction temperature to 160 °C (Table 2, pathway I, yields in parentheses).

The synthesis is strong with regard to unsymmetrically substituted hydro-1H-carbazoles (Table 2, pathway I, 1b and 1c). The dehydrogenation of the octahydrocarbazole intermediates required a reaction temperature of 190 °C and gave an almost quantitative yield for the dehydrogenation step (Table 3, pathway I). With the high yields observed for the hydrogenation (Table 1, entries 1 and 8) and the dehydrogenation step (Table 3, pathway I), the ADC step determines the overall yield. 3-Methylcarbazole is a common precursor for carbazole alkaloids in plants[22] and could be isolated in nearly 70 % overall yield. The formation of quinolines or acridines becomes feasible by applying 1,3-aminoalcohols or (2-aminophenyl)methanols, respectively, instead of aminophenols. The reaction conditions for the ADC reaction step were optimized at first. To our delight, the reusable Ir@SiCN catalyst was highly efficient. A somewhat higher catalyst loading than that used for the carbazole synthesis and a reaction temperature of 140 °C were necessary to mediate the ADC step. The functionalization of phenol with various 1,3-aminoalcohol components resulted in the formation of tetrahydroquinolines in up to 77% isolated yield (Table 2, pathway II, 2d). Dehydrogenation of the tetrahydroquinolines proceeds with 88 to 97 % isolated yields. Again, the ADC step determines the overall yield. The utilization of (2-aminophenyl)methanols resulted in the formation of acridines (Tables 2 and 3, pathway III). Yields up to 88% have been observed for the ADC step (Table 2, pathway III). A variety of unsymmetrically alkylated and arylated tetrahydroacridines has been synthesized (Table 2, pathway III, 3b-3f). The dehydrogenation of tetrahydroacridines proceeds quantitatively (Table 3, pathway III). To demonstrate that hydrogen is indeed released in the ADC steps and the dehydrogenation steps, the released hydrogen was quantified for the reaction of cyclohexanol with 2-aminobenzyl alcohol (ADC) and for the associated dehydrogenation of 1,2,3,4-tetrahydroacridine. The results are in good agreement with the theoretically expected values (Supplementary Table 6,7).

**Reaction sequence without isolation of intermediates**. Since it is possible to apply reusable catalysts for all of the reaction steps, the overall reaction sequence can be performed without isolating the intermediate products (Fig. 3). The hydrogenation of phenol is now performed in tetrahydrofuran, since the presence of water is detrimental to the second step involving condensation reactions. The Ru@SiCN catalyst was separated by centrifugation and a mixture of Ir@SiCN catalyst, KO$^t$Bu, diglyme and 2-aminobenzyl alcohol was added to the cyclohexanol solution. The mixture was evacuated and flushed with argon three times and stirred at 140 °C for 22 h. Water was added to more easily remove the ADC catalyst (Ir@SiCN). The Pd@SiCN catalyst was added to the organic phase and the low boiling solvents were evaporated. The acceptorless dehydrogenation at elevated temperature finally yielded the acridine product in overall yields between 79 and 84%. The catalysts for all reaction steps were purified by centrifugation and the entire procedure was repeated three times to demonstrate the reusability of the Ru@SiCN, Ir@SiCN and Pd@SiCN catalysts (Fig. 3).

**Discussion**

We expect the catalytic condensation introduced here to become an integral part of the new alcohol based sustainable chemistry that permits the catalytic synthesis of organic compounds from aliphatic and/or aromatic alcohols. Mechanistically, our overall catalytic condensation takes place in three steps: hydrogenation, dehydrogenative condensation and dehydrogenation. In the dehydrogenative condensation and in the dehydrogenation steps, hydrogen is liberated nearly quantitatively as demonstrated for

---

**Table 1 Hydrogenation of phenolic compounds[a]**

| Entry | R | Yield (%)[b] |
|---|---|---|
| 1[c] | None | >99 |
| 2[d] | None | 97[e] |
| 3 | 1-methyl | >99 |
| 4 | 1-ethyl | >99 |
| 5 | 4-methyl | >99 |
| 6 | 4-tert-butyl | >99 |
| 7 | 3,5-dimethyl | 92 |
| 8[f] | 2-amino | 98 |

[a] 1 mmol substrate, 50 °C, p(H₂) = 20 bar, 5 mg Ru@SiCN catalyst (0.03 mol% active metal), 1 mL water, 20 h
[b] Yields determined by GC and GC-MS using dodecane as internal standard
[c] 50 °C, 3 bar H₂ pressure, 24 h
[d] 100 mmol substrate, 50 °C, p(H₂) = 20 bar, 200 mg Ru@SiCN catalyst (0.01 mol% active metal), 10 mL water, 24 h. The reactor was pressured again to 20 bar after half of the reaction time
[e] Yield of isolated product
[f] 80 °C, p(H₂) = 50 bar, 24 h, 20 mg catalyst (0.12 mol% active Ru)

**Table 2 Synthesis of heterocyclic compounds via ADC[a]**

| Pathway | Product | | Yield[b] [%] |
|---|---|---|---|
| (I) | 1a: R = H | | 81 (79) |
| | 1b: R = 3-methyl | | 70 (65) |
| | 1c | | 53 (56) |
| (II) | 2a: R = H; R' = H | | 58 |
| | 2b: R = H; R' = $C_{11}H_{23}$ | | 72 |
| | 2c: R = H; R' = p-tolyl | | 72 |
| | 2d: R = H; R' = 3,4-dimethoxyphenyl | | 77 |
| | 2e: R = H; R' = pyridine-3-yl | | 62 |
| | 2f: R = H; R' = 4-chlorophenyl | | 68 |
| | 2g: R = H; R' = 4-bromophenyl | | 61 |
| (III) | 3a: R = H; R' = H | | 79 |
| | 3b: R = 2-*tert*-butyl; R' = H | | 87 |
| | 3c: R = 2-methyl; R' = H | | 68 |
| | 3d: R = 4-methyl; R' = H | | 76 |
| | 3e: R = H; R' = 7-Cl | | 72 |
| | 3f: R = H | | 88 |
| | 3g: R = methoxy | | 79 |

[a] Reaction conditions: (I): 2.0 mL catalyst **I** (0.02 mmol, 0.01 M in thf), cyclohexanol compound (15.22 mmol), 2-aminocyclohexanol (7.61 mmol), 10 mL thf, 1.1 eq. KO$^t$Bu, 105 °C, 22 h; Yields in parentheses are given for the heterogeneous catalyst mediated reaction at 160 °C. (II)(III): 150 mg Ir@SiCN (0.5 mol% active metal), cyclohexanol compound (12.0 mmol), 1,3-aminoalcohol (3.0 mmol), 3 mL diglyme, 2.0 eq. KO$^t$Bu, 140 °C, 24 h
[b] Yields of isolated products

**Table 3 Acceptorless dehydrogenation by Pd@SiCN[a]**

| Pathway | Product | | Yield[b] [%] |
|---|---|---|---|
| (I) | (carbazole) | 4a | >99 |
| | (methyl carbazole) | 4b | 97 |
| | (benzo-carbazole) | 4c | 96 |
| (II) | (quinoline, R–N–R') | 5a: R = H; R' = H | 92 |
| | | 5b: R = H; R' = C$_{11}$H$_{23}$ | 88 |
| | | 5c: R = H; R' = p-tolyl | 94 |
| | | 5d: R = H; R' = 3,4-dimethoxyphenyl | 93 |
| | | 5e: R = H; R' = pyridine-3-yl | 97 |
| (III) | (acridine) | 6a: R = H | 97 |
| | | 6b: R = 2-*tert*-butyl | 98 |
| | | 6c: R = 2-methyl | 96 |
| | | 6d: R = 4-methyl | 93 |
| | (benzacridine) | 6e: R = H | 98 |
| | | 6f: R = methoxy | 98 |

[a] Reaction conditions: 50 mg (0.18 mol% active metal) Pd@SiCN, 1.0 mmol substrate, 0.75 mL diglyme, 180–200 °C, 18 h (under slight argon flow)
[b] Yields of isolated products

one example of each step. The formation of C–C bonds from aromatic carbon atoms formally provided by catalytic C–H bond activations and C–OH bond hydrogenolysis is a unique feature of our novel synthesis concept. Analogously, C–N bond formation

takes place. We have exemplarily demonstrated how carbazoles, quinolines and acridines may readily built up by catalytic condensation. More polycyclic heteroaromatic patterns should be within reach. The reaction sequences discussed here can also start

**Fig. 3** Direct synthesis of acridine and reusability of the catalysts. **a** Hydrogenation by Ru@SiCN and catalyst removal. **b** ADC step applying Ir@SiCN and catalyst removal. **c** Dehydrogenation using Pd@SiCN and work up of the products. The synthesis was repeated for three times with the same catalysts. The overall yields of the isolated acridine vary between 79 and 84%

from cyclohexanol derivatives instead of phenols. Cyclohexanol derivatives are formed in many lignin hydrogenolysis reactions since the phenol building blocks of this biopolymer are hydrogenated simultaneously[20]. Since our catalytic condensation reaction can form completely aromatic compounds, the concept is an option to regain aromatic moieties from hydrogenated cyclic educts. In alcohol activation concepts, which have been developed so far, such as borrowing hydrogen[23] or hydrogen auto-transfer[24] and ADC[3,26], the alcohol activation takes place via dehydrogenation[25]. We activated alcohols via hydrogenation in the catalytic condensation reactions introduced here.

## Methods

**Direct synthesis of acridine**. 12 mmol of phenol, 25 mg (0.096 mol% active metal) Ru@SiCN, 1.5 mL THF, p(H$_2$) = 20 bar, $T$ = 50 °C, 24 h were stirred overnight. The catalyst was removed by centrifugation and the supernatant solution was added to 3 mmol 2-aminobenzyl alcohol, 6 mmol KO$^t$Bu, 3 mL diglylme, 150 mg (0.50 mol% active metal) Ir@SiCN, $T$ = 140 °C, 20 h. Catalyst was removed by addition of water and the organic phase was collected. After adding the mixture to 100 mg (0.12 mol% active metal) Pd@SiCN, the solvents were removed under reduced pressure and then heated at 190 °C for 36 h at an slight Ar flow of 4–6 mL/min.

**Data availability**. All data are available from the authors upon reasonable request.

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

## Acknowledgements
We acknowledge financial support from the Deutsche Forschungsgemeinschaft, KE 756/23-2.

## Author contributions
D.F. and R.K. designed the experiments. D.F. and T.S. carried out the synthesis experiments (catalysts and organic compounds) and analyzed the spectroscopic data. All authors wrote the manuscript.

## Additional information

**Competing interests:** The authors declare no competing interests.

