## [Peer Review File · Nature Communications]

Reviewers' comments:

Reviewer #1 (Remarks to the Author):

In this contribution, Kempe and co-workers impressively demonstrate the power of alcohol based sustainable chemistry that permits the catalytic synthesis of organic compounds from aliphatic and/or aromatic alcohols. The authors exemplarily show the sustainable syntheses of several carbazoles, quinolines and acridines which are readily built up by catalytic condensation reactions. As catalysts reusable supported Ru, Pd and Ir catalysts (Ru@SiCN, etc.) were used. All experiments are competently carried out. This manuscript is very carefully and clearly written, and thus enjoyable to read. There is nothing to criticize and in my opinion the chemistry reported will be of great interest to the readers of Nature Commun. This paper strongly highlights the huge potential of metal catalysts for possible future applications in biomass conversion. I recommend this contribution for publication.

Reviewer #2 (Remarks to the Author):

Sustainable conversion of biomass-derived alcohols into valuable chemicals is highly desirable but a challenging task. This paper described a novel strategy for an atom-economic synthesis of useful polycyclic heteroaromatic compounds from readily available phenols and amino alcohols. The reactions based on this concept proceeds via a catalytic hydrogenation, condensation and dehydrogenation sequence. Notably, in this reaction process the mass balance for H₂ is perfect for many cases without loss or input of H₂ theoretically. More importantly, three different reusable heterogeneous catalysts developed previously by the same group could be utilized in these continuous three steps without the isolation of any intermediates. It well demonstrates the application potential of this synthetic methodology, although the functional group compatibility is still needed to be improved in the future work. Therefore, I support the publication of this nice work on Nature commun. with minor revisions.

The comments and suggestions are as following:

1. It is only a pity that the authors used homogeneous Ir catalyst I for condensation reaction towards carbazole synthesis and mentioned that it is more efficient than the Ir@SiCN catalyst. Anyway, I suggest the authors list the results using Ir@SiCN catalyst for this step in Table 2. Although this catalyst is less active, these results can be a good proof of concept for this novel

sustainable synthetic method. Meanwhile, it is interesting as well to make a comparison between homogeneous and heterogeneous Ir catalysts for this reaction. Please also show the structure of Ir catalyst I in Table 2.

2. The mass balance of H₂ is very impressive for this designed reaction sequence. Therefore, I suggest the authors to measure the yield of H₂ produced for one or two examples in Table 2. The result will well demonstrate the atom-economic of this strategy. A GC analysis of the gas phase to show the constituents is also informative for the selectivity of dehydrogenation process.

3. In SI part, ¹H NMR scales of 2e, 4c, 5a, 5e and 6a should range from 0 to 10 ppm; On page S12, Compound 3d should be further purified to obtain the clean NMR spectra; On page S19, integral of aromatic C-H at 7.5 ppm for ¹H NMR of Compound 5d is missing.

Reviewer #3 (Remarks to the Author):

In this manuscript the authors present their approach for the synthesis of heteroaromatics such as carbazoles, quinolines and acridines and highlight its importance in the context of biomass utilisation and CO₂-emission reduction strategies. The main claim is that they have developed a new catalytic condensation reaction of phenols and aminophenols for the synthesis of heteroaromatics.

While the results obtained in this work can be useful for the synthesis of the highlighted heteroaromatics (although the scope should be expanded, since nothing has been demonstrated or mentioned about more interesting, functionalised heteroaromatics), I find the way the work is presented, and claims are made unclear and even misleading.

For example, the authors write about “catalytic condensation introduced here”, and “Mechanistically, synchronized catalytic hydrogenation, multiple condensation, and dehydrogenations ...”, but I cannot find any basis for these claims. I carefully read the manuscript twice, trying to find the corresponding results, but did not succeed.

What the authors actually present here is an optimisation of three different reactions (that on their own are not conceptually new) and a demonstration of one example in which the optimised reactions are performed consecutively without purification of the intermediate products. However, it should be noted that in the latter example additional operations are required for catalyst separation (figure below).

Therefore, this cannot even be considered as one pot synthesis. The first reaction is a well established hydrogenation of phenols and for this step the authors optimise the reaction for their own catalyst. Then they optimise the next step, which is the so called ADC that the authors presented earlier in their Nature Chemistry paper for the synthesis of pyrroles (cited in this work as well). Finally they optimise another well-established reaction, namely dehydrogenation, for

their catalyst.

Thus, in my opinion the main novelty of this paper is the application of the authors' previously reported method for pyrrole synthesis to other heteroaromatics, such as carbazoles, quinolines and acridines. This is interesting in its own right and can be considered for publication in Nature Communications, but not before:

- 1) the paper is rewritten substantially to avoid all the misleading claims,
- 2) Synthesis of more elaborated heteroaromatics (2-3 examples) with functional groups is presented,
- 3) the claim regarding the importance of their methodology for biomass valorisation is supported by applying this approach directly to the mixture of aromatic products derived from lignin.

Looking at the experimental procedure I doubt this is possible, since the chemistry seems very sensitive. Of course, if this is not possible then these claims must be removed.

Reviewers comments "Remarks to the Author" are copied below, our comments start with "Our response:" (blue), and the description of what we have changed start with "Our alteration:" (red). The changes in the manuscript based on the issues raised by the reviewers are marked with a yellow background.

Reviewer #1:

"In this contribution, Kempe and co-workers impressively demonstrate the power of alcohol based sustainable chemistry that permits the catalytic synthesis of organic compounds from aliphatic and/or aromatic alcohols. The authors exemplarily show the sustainable syntheses of several carbazoles, quinolines and acridines which are readily built up by catalytic condensation reactions. As catalysts reusable supported Ru, Pd and Ir catalysts (Ru@SiCN, etc.) were used. All experiments are competently carried out. This manuscript is very carefully and clearly written, and thus enjoyable to read. There is nothing to criticize and in my opinion the chemistry reported will be of great interest to the readers of Nature Commun. This paper strongly highlights the huge potential of metal catalysts for possible future applications in biomass conversion. I recommend this contribution for publication."

"Our response": Thanks to Reviewer #1 for evaluating our manuscript!

Reviewer #2:

"Sustainable conversion of biomass-derived alcohols into valuable chemicals is highly desirable but a challenging task. This paper described a novel strategy for an atom-economic synthesis of useful polycyclic heteroaromatic compounds from readily available phenols and amino alcohols. The reactions based on this concept proceeds via a catalytic hydrogenation, condensation and dehydrogenation sequence. Notably, in this reaction process the mass balance for H₂ is perfect for many cases without loss or input of H₂ theoretically. More importantly, three different reusable heterogeneous catalysts developed previously by the same group could be utilized in these continuous three steps without the isolation of any intermediates. It well demonstrates the application potential of this synthetic methodology, although the functional group compatibility is still needed to be improved in the future work. Therefore, I support the publication of this nice work on Nature commun. with minor revisions."

"Our response": Thanks to Reviewer #2 for evaluating and improving our manuscript!

"The comments and suggestions are as following:

1.It is only a pity that the authors used homogeneous Ir catalyst I for condensation reaction towards carbazole synthesis and mentioned that it is more efficient than the Ir@SiCN catalyst. Anyway, I suggest the authors list the results using Ir@SiCN catalyst for this step in Table 2. Although this catalyst is less active, these results can be a good proof of concept for this novel sustainable synthetic method. Meanwhile, it is interesting as well to make a comparison between homogeneous and heterogeneous Ir catalysts for this reaction. Please also show the structure of Ir catalyst I in Table 2."

“Our response”: Thanks, very good suggestions! In addition, we repeated the reactions with the heterogeneous catalyst to see if we get a better conversion at more elevated temperature. At 160 °C instead of 140 °C, we now see product formation of the heterogeneous catalyst similar to that of the homogeneous catalyst.

“Our alteration”: We list the results for the Ir@SiCN catalyst (carbazole synthesis step at 160 °C) in Table 2 to make a comparison between the homogeneous and the heterogeneous Ir catalysts. In addition, structure of the Ir catalyst I was added to Table 2. Details of the experiments including the originally 140 °C runs were added to the SI.

“2.The mass balance of H₂ is very impressive for this designed reaction sequence. Therefore, I suggest the authors to measure the yield of H₂ produced for one or two examples in Table 2. The result will well demonstrate the atom-economic of this strategy. A GC analysis of the gas phase to show the constituents is also informative for the selectivity of dehydrogenation process.”

“Our response”: Thanks, very good suggestion! We quantified the amount of liberated H₂ for the reaction of cyclohexanol and 2-aminobenzyl alcohol (ADC step) and the associated dehydrogenation of 1,2,3,4-tetrahydroacridine. The results are well in accordance with the theoretically expected release.

“Our alteration”: A short comment was added to the manuscript text explaining these experiments/results. Details of the experiments were added to the SI.

“3.In SI part, ¹H NMR scales of 2e, 4c ,5a, 5e and 6a should range from 0 to 10 ppm; On page S12, Compound 3d should be further purified to obtain the clean NMR spectra; On page S19, integral of aromatic C-H at 7.5 ppm for ¹H NMR of Compound 5d is missing.”

“Our response”: Thanks! The ¹H NMR of **2e**, **4c**, **5a**, **5e** and **8a** were plotted from 1-10 ppm. In addition, we further repeated the synthesis of **3d** to obtain a pure material. In addition, the missing integral for **5b** was added.

“Our alteration”: The corresponding details were added to the SI!

Reviewer #3:

“In this manuscript the authors present their approach for the synthesis of heteroaromatics such as carbazoles, quinolines and acridines and highlight its importance in the context of biomass utilisation and CO₂-emission reduction strategies.”

“Our response”: Thanks to Reviewer #3 for evaluating and improving our manuscript!

The main claim is that they have developed a new catalytic condensation reaction of phenols and aminophenols for the synthesis of heteroaromatics.

While the results obtained in this work can be useful for the synthesis of the highlighted heteroaromatics (although the scope should be expanded, since nothing has been demonstrated or mentioned about more interesting, functionalised heteroaromatics), I find the way the work is presented, and claims are made unclear and even misleading.”

“Our response”: Thanks, very important points! We added three more examples with functional groups (ADC step). For details, see below point #2 of this reviewer. Furthermore, we either added experiments to support claims, or explained in more detail what we mean to avoid misleading. For details, see point #1 of this reviewer.

“Our alteration”: See below points #1 and 2 of this reviewer!

For example, the authors write about “catalytic condensation introduced here”, and “Mechanistically, synchronized catalytic hydrogenation, multiple condensation, and dehydrogenations ...”, but I cannot find any basis for these claims. I carefully read the manuscript twice, trying to find the corresponding results, but did not succeed.”

“Our response”: We call the overall reaction a condensation since water is produced as the by-product. The overall carbazole synthesis is a good example since water is the only byproduct. We add six H₂ molecules in the hydrogenation step (Table 1) and liberate six H₂ molecules in the ADC and the dehydrogenation steps. I fully agree; we did not demonstrate that hydrogen is quantitatively released in the ADC steps and the dehydrogenation steps. Now, we also quantify the released amount of hydrogen, one example for each step (ADC and dehydrogenation), and see a good agreement of theoretical and measured H₂ release. For details, see point #2 of reviewer 2.

“Our alteration”: We now explain better what we mean with catalytic condensation and change the sentence “Mechanistically, synchronized catalytic hydrogenation, multiple condensation, and dehydrogenations steps lead to the selective formation of crucial C-C and C-N bonds.” Please see point #1 of this reviewer for details.

“What the authors actually present here is an optimisation of three different reactions (that on their own are not conceptually new) and a demonstration of one example in which the optimised reactions are performed consecutively without purification of the intermediate products. However, it should be noted that in the latter example additional operations are required for catalyst separation (figure below). Therefore, this cannot even be considered as one pot synthesis. The first reaction is a well established hydrogenation of phenols and for this step the authors optimise the reaction for their own catalyst. Then they optimise the next step, which is the so called ADC that the authors presented earlier in their Nature Chemistry paper for the synthesis of pyrroles (cited in this work as well). Finally they optimise another well-established reaction, namely dehydrogenation, for their catalyst. Thus, in my opinion the main novelty of this paper is the application of the authors’ previously reported method for pyrrole synthesis to other heteroaromatics, such as carbazoles, quinolines and acridines.

This is interesting in its own right and can be considered for publication in Nature Communications, but not before.”

“Our response”: Thanks! We fully agree!

“1) the paper is rewritten substantially to avoid all the misleading claims,”

“Our response”: Thanks! We went through all the claims that could be misleading and altered them. In addition, text was changed based on work suggested by this and other reviewers.

“Our alteration”:

Abstract:

Old: The reactions of this synthesis concept proceed via hydrogenation and multiple dehydrogenation–condensation steps.

New: The overall reaction of this synthesis concept proceed via three steps: hydrogenation, dehydrogenative condensation and dehydrogenation.

The reusable catalyst claim is fully valid now since we also demonstrate the use of reusable catalyst for the carbazole ADC step.

Introduction:

Old: The concept is also suitable to meet the challenges associated with aryl ether hydrogenolysis, a key step of lignin valorization²⁰.

New: The concept might be suitable to meet some challenges associated with aryl ether hydrogenolysis, a key step of lignin valorization²⁰.

We also deleted the following sentence since our intention is better explained in the conclusion section. “Unfortunately, lignin is the most promising sustainable source of aromatic compounds. Aromatic moieties can be recovered by feeding the hydrogenated phenols into the reactions discussed herein.”

Results and Discussion:

Old: Since it is possible to apply reusable catalysts for most of the reaction steps,

New: Since it is possible to apply reusable catalysts for all of the reaction steps,

Conclusion:

Old: Mechanistically, synchronized catalytic hydrogenation, multiple condensation, and dehydrogenations steps lead to the selective formation of crucial C-C and C-N bonds.

New: Mechanistically, our overall catalytic condensation takes place in three steps: hydrogenation, dehydrogenative condensation and dehydrogenation. In the dehydrogenative condensation and in the dehydrogenation steps, hydrogen is liberated nearly quantitatively as demonstrated for one example of each step.

Old: This variation permits the use the unselective lignin hydrogenolysis products, which are formed by most of the available catalysts.

New: Cyclohexanol derivatives are formed in many lignin hydrogenolysis reactions since the phenol building blocks of this biopolymer are hydrogenated simultaneously²⁰.

We also delete “In such reactions, the liberation of up to multiple equivalents of H₂ is observed.” Since we discuss the hydrogen liberation now earlier in the manuscript.

Old: Since our catalytic condensation reaction forms completely aromatic compounds, the concept is an option to regain aromatic moieties from hydrogenated lignin cleavage products.

New: Since our catalytic condensation reaction can form completely aromatic compounds, the concept is an option to regain aromatic moieties from hydrogenated cyclic educts.

“2) Synthesis of more elaborated hereroaromatics (2-3 examples) with functional groups is presented,”

“Our response”: Thanks, very good suggestion! We additionally synthesized 2-(4-chlorophenyl)-5,6,7,8-tetrahydroquinoline, 2-(4-bromophenyl)-5,6,7,8-tetrahydroquinoline and 7-chloro-1,2,3,4-tetrahydroacridine (ADC step) to have more examples with functional groups.

“Our alteration”: We added the examples to Table 2 and the experimental details to the SI.

“3) the claim regarding the importance of their methodology for biomass valorisation is supported by applying this approach directly to the mixture of aromatic products derived from lignin. Looking at the experimental procedure I doubt this is possible, since the chemistry seems very sensitive. Of course, if this is not possible then these claims must be removed.”

“Our response”: We fully agree with the reviewer that we are not able to use real biomass-valorization product mixtures. We are actually interested in developing reactions in which alcohols are converted into important classes of chemical compounds because alcohols can be obtained from biomasses such as lignocellulose see, for instance, ref 2 or other processes like CO₂ hydrogenation or sugar fermentation. To use alcohols in our reaction, they have to be purified and/or partially converted, for instance, diols have to be reacted with amines/ammonia to synthesize amino alcohols.

“Our alteration”: We again went through the manuscript to make sure that we do not write we are able to use biomass-valorization product mixtures directly.

Reviewers' Comments:

Reviewer #2 (Remarks to the Author):

The authors carefully revised the manuscript according to the referee report. I think all the valid points from the referees have been well addressed. Therefore, I fully support the publication of this revised version on nature commun.

Reviewer #3 (Remarks to the Author):

I am satisfied with revisions. Content-wise the manuscript can be accepted for the publication.